

# Brief communication: Identification of 140,000-year-old blue ice in Grove Mountains, East Antarctica, by krypton-81 dating

Zhengyi Hu[a], Wei Jiang[b,c], Yuzhen Yan[d], Yan Huang[e], Xueyuan Tang[a], Lin Li[a], Florian
Ritterbusch[b,c], Guo-Min Yang[b,c], Zheng-Tian Lu[b,c], Guitao Shi[e,f*]

[a] MNR Key Laboratory for Polar Science, Polar Research Institute of China, Shanghai, 200062, China

[b] Hefei National Laboratory, University of Science and Technology of China, Hefei, 230088, China

[c] Hefei National Research Center for Physical Sciences at the Microscale and School of Physical Sciences, University of Science and Technology of China, Hefei, 230026, China

[d] State Key Laboratory of Marine Geology, Tongji University, Shanghai, 200092, China

[e] Key Laboratory of Geographic Information Science, School of Geographic Sciences, East China Normal University, Shanghai, 200241, China

[f] State Key Laboratory of Estuarine and Coastal Research, East China Normal University, Shanghai 200241, China

*Correspondence to: GS, gtshi@geo.ecnu.edu.cn

**Abstract: (<100 words)**

The presence of exceptionally old ice and the relative ease of access make Antarctic blue ice areas (BIAs) attractive paleoclimate archives. However, only a handful of BIAs, mostly situated in West Antarctica and along the Trans-Antarctic Mountains, have been investigated for this purpose. Here, we present the age of surface ice from the Grove Mountains BIA in

Elizabeth Princess Land, East Antarctica, determined by measuring $^{81}$Kr in the trapped air. Two samples yield an average age of $143^{+33}_{-29}$ kyr. Together with the reported terrestrial age of a chondrite, we conclude that Grove Mountains BIA holds considerable potential for paleoclimate studies.




## 1. Introduction

Antarctic ice cores provide a wealth of information about the Earth's past climate and atmospheric composition, especially greenhouse gases (e.g., Petit et al., 1999). International efforts are underway to locate and retrieve an ice core dating back to 1.5 Myr that is in stratigraphic order (Fischer et al., 2013). However, such endeavors are expensive and time-consuming. Shallow drilling in blue ice areas (BIAs) in Antarctica has therefore emerged as a complementary approach (Yan et al., 2019). BIAs are regions where ablation exceeds accumulation. The negative mass balance at the surface is maintained by the supplies of crystalline glacial ice from below. In this case, ancient ice that was once buried deep in the ice sheet is being exhumed and can be readily accessed. The presence of meteorites that have terrestrial ages up to 2 Myr in the BIAs hints at the existence of ice that is older than 800 kyr (Scherer et al., 1997).

In Antarctica, a number of BIAs have been investigated to understand local meteorology, glaciology, and meteorites (e.g., Liu et al., 2010; Scherer et al., 1997; Spaulding et al., 2013). Nevertheless, to date debris-free ice samples have been recovered from only five blue ice areas for the purpose of paleoclimate studies (Table 1 and Figure 1): Mount Moulton and Patriot Hills in West Antarctica, and Allan Hills, Taylor Glacier, and Larsen Glacier in East Antarctica. All three East Antarctic BIAs are situated in Victoria Land despite considerable presence of BIAs in other parts of East Antarctica, such as near Grove Mountains. The Grove Mountains, consisting of a series of nunataks southwest of Princess Elizabeth Land, East Antarctica approximately 400 km from the Antarctic coast, is on a different side of the continent, far away from the cluster of previous sites (Figure 1). Here, we report the age of surface ice in Grove Mountains determined by $^{81}$Kr dating and evaluate the potentials of Grove Mountains BIA as paleoclimate archives.

**Table 1. List of Antarctic blue ice areas as paleoclimate archives**

| Areas | Location | Age range | References |
|---|---|---|---|
| Grove Mountains | 72.99° S, 75.22°E | 143 kyr | This study |
| Mount Moulton | 76.7° S, 134.7° W | 105-136 kyr | (Korotkikh et al., 2011) |
| Patriot Hills | 80.3° S, 81.4° W | 10-80 kyr;130-134 kyr | (Turney et al., 2020) |
| Allan Hills | 76.7° S, 159.4° E | 90-250 kyr; >1 Myr | (Spaulding et al., 2013; Yan et al., 2019) |
| Taylor Glacier | 77.8° S, 161.8° E | 9-133 kyr | (Buizert et al., 2014) |
| Larsen Glacier | 74.9° S, 161.6° E | 9-25 kyr | (Lee et al., 2022) |



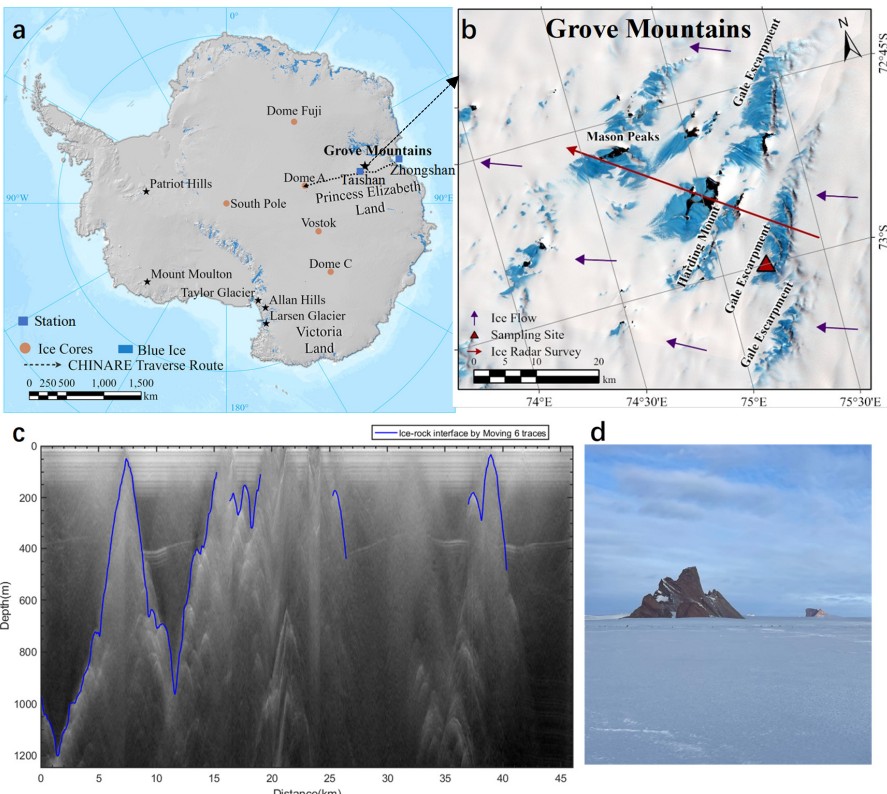

**Figure 1 Antarctic blue ice areas (BIAs) and the Grove Mountains. (a)** Sites of BIAs explored as paleoclimate archives listed in Table 1 (closed black stars), sites of deep ice coring, and the traverse route of this work from coastal Zhongshan station to Dome A (base map from ESRI). **(b)** Satellite imagery of the Grove Mountains BIAs with local ice flow lines (purple arrows) and the transect of an airborne ice radar survey (red arrow) (base image is Landsat Image Mosaic of Antarctica). The sampling site for this study is marked with a red triangle. **(c)** Radar profile of the transect. **(d)** The uneven surface of blue ice in Grove Mountains and protruding nunataks are similar to the Allan Hills BIAs described by Spaulding et al. (2013).

## 2. Materials and methods

### 2.1 Site and sample description

The Grove Mountains BIA is located in the southwest of the Princess Elizabeth Land, East Antarctica (Figure 1). Ice flows from the southeast to the northwest and eventually drains into the Lambert Glacier. Ice flow is blocked by a series of nunataks in this region. The combined





effect of glacial flows and topographic barriers leads to the formation of blue ice as well as a large number of crevasses that prevent access for ice drilling. The highest nunatak, Mason Peak, has an elevation of 2365 m and a topographic prominence of ~700 m. The average ice elevation

of Grove Mountains is ~2000 m, and past radar surveys revealed steep bedrock topography and thick (>1000 m) ice layers (Figure 1). The annual average temperature in Grove Mountains is ~-30 ºC.

Similar to other Antarctic BIAs, Grove Mountains BIA is a meteorite concentration site. To date, more than 10000 meteorites have been recovered by the Chinese National Antarctica

Research Expedition (CHINARE) from this region. In January 2016, surface blue ice (up to ~50 cm in depth) was collected manually using stainless-steel spades near the mid-Gale Escarpment where the meteorites are concentrated. In total, about 40 kg of ice was collected (72.99º S, 75.22º E). A ~5 cm thick layer of surface ice was removed before sampling. The collected ice was kept in clean polyethylene bags and then preserved in insulated cabinets and

transported under freezing conditions (-20 ºC).

**2.2 Analytical methods**

The blue ice collected from Grove Mountains was processed in two batches. The first batch (batch A) contains large ice blocks (size ~20 cm or bigger) with a total weight of 26 kg. The

second batch (batch B) contains small ice pieces (size 10-20 cm) and weighs about 9 kg. The reason to separate them into two batches is to see if there are more modern air contaminations in the batch with small ice samples. Before melting, the outer 3-5 mm layer of the ice was removed to reduce potential contamination from modern air. The ice was then melted in an evacuated container. The released air was transferred into 1L stainless steel bottles with a

compressor. The extraction system typically achieves recoveries > 95% and contamination < 1 ‰. The gas contents obtained for sample batch A and batch B were 95 mL STP/kg and 86 mL STP/kg, respectively. The difference is probably due to the larger specific surface area of the smaller sample (batch B) compared to the large one (batch A) and hence greater gas losses from smaller ice samples. Nonetheless, these gas contents are comparable to the blue ice

retrieved from other BIAs in Antarctica (e.g., Buizert et al., 2014). Krypton (Kr) was separated from the extracted air with a purification system based on titanium gettering and gas chromatography (Jiang et al., 2020), yielding Kr purities and recoveries higher than 90%. 2.2 µL STP and 0.6 µL STP Kr were obtained for the larger and smaller sample, respectively.

The Kr sample was measured with the atom trap trace analysis (ATTA) method (Jiang et al.,

2020). $^{81}$Kr and $^{85}$Kr atoms are selectively captured by laser beams into a magneto-optical trap



and are counted by detecting their fluorescence. Meanwhile, $^{83}$Kr (a stable isotope) is chosen as the reference and its capture rate is also measured. Each analysis took about 5 hours. The measurement is cycled between the atom-counting mode for $^{81}$Kr and $^{85}$Kr, and the capture-rate mode for $^{83}$Kr in order to cancel the slow drifts in the capture efficiencies of the instrument.

As the amount of Kr from the ice samples is generally limited, it is important to keep the cross-sample contamination under control. This effect comes from the discharge source in the ATTA instrument. On the one hand, it slowly consumes the Kr sample in the system and makes the effective sample size smaller. On the other hand, Kr from the previous measurement is slowly released from vacuum parts to cause cross-sample contamination. In order to reduce this effect,

the vacuum system of the ATTA instrument was washed continuously with a Xe discharge for one week. After the washing, the Kr outgassing rate was about $10^{-3}$ μL STP/h. For a 5-hour measurement the cross-sample contamination is 1~3%. The residual cross-sample contamination effect was corrected based on the $^{85}$Kr measurement.

The measured relative $^{81}$Kr abundance was used to calculate the age of the sample based on

the radioactive decay law and the atmospheric input function of $^{81}$Kr. The uncertainty of the $^{81}$Kr age mainly came from the statistical errors for atom counting. The uncertainty caused by the cross-sample contamination correction was included through error propagation. Since the atom counts for both $^{81}$Kr and $^{85}$Kr were low (10 ~ 100), we adopted the Feldman-Cousins method, which provided a unified approach to treat measurements with small signals (Feldman

and Cousins, 1998). Besides the statistical error, there were additional systematic errors due to the uncertainty of the half-life of $^{81}$Kr (229 ±11 ka) and the uncertainty of the atmospheric $^{81}$Kr input functions (Zappala et al., 2020). Note that the measurements were performed in 2017. The ATTA instrument has been improved significantly since then. The sample requirement is now less than 2.0 kg and the dating precision is also better (Crotti et al., 2021).

Two pieces of blue ice from batch A were randomly selected for chemical measurements. In a Class 1000 clean room, the surface layer of ~1 cm was washed with ultrapure Milli-Q water (18.2MΩ) to remove any surface contaminants. Then, the ice was melted under a super clean hood (Class 100) at 20ºC for chemical measurements. The major chemical ions, $Na^+$, $K^+$, $Mg^{2+}$, $Cl^-$, $NO_3^-$, and $SO_4^{2-}$, were determined with an ICS-3000 IC system (Dionex, USA). More

details on ion analysis are provided in Shi et al. (2012). The $\delta^{18}$O and $\delta$D of ice were measured with a wavelength scanned cavity ring down spectroscopy (WS-CRDS) instrument, Picarro L-2130i (Picarro Inc., USA), with the respective analysis precision of 0.05‰ and 0.5‰. Details on the water isotope analysis are described in a previous study (Ma et al., 2020).




## 3. Results

The results of radiometric Kr dating of two batches of ice from the Grove Mountains BIAs are shown in Table 2. The isotopic abundances of [81]Kr in the trapped air of the two batches are statistically indistinguishable from each other. Furthermore, despite the different ice sizes and weights, both batches show non-zero but similar level of [85]Kr activities, indicating modern air contaminations possibly due to cracks near the blue ice surface. Similar intrusion of the modern atmosphere to the blue ice has previously been observed in other BIAs as well (Spaulding et al., 2013). After correcting for the modern air contamination (assuming an atmospheric [85]Kr activity of 70 ± 5 decay per minute per cubic centimeter krypton at STP (dpm/cc) at the sampling time (25 Jan. 2016); Kersting et al. (2020)), the averaged relative [81]Kr/Kr is 65 ± 6 (pMKr), which corresponds to an age of $143^{+33}_{-29}$ kyr. This age is comparable to the previously reported age of the surface ice in Mount Moulton, Patriot Hills and Allan Hills BIAs (Korotkikh et al., 2011; Spaulding et al., 2013; Turney et al., 2020). Moreover, a CR chondrite (No. GRV 021710) discovered near the sampling site has a terrestrial age of 260 kyr (Lu, 2008). These results hint at the presence of even older blue ice in the vicinity of Grove Mountains.

**Table 2 [81]Kr dating of blue ice near Grove Mountains**

|  | Sample size (kg) | Kr extracted (µL STP) | [85]Kr activity at sampling time (dpm/cc) | [81]Kr/Kr (pMKr) | [81]Kr/Kr_corrected (pMKr) | [81]Kr age (ka) |
|---|---|---|---|---|---|---|
| **Batch A** | 26 | 2.2 | 27 ± 1 | 76 ± 5 | 61 ± 8 | $165^{+48}_{-43}$ |
| **Batch B** | 9 | 0.7 | 24 ± 1 | 82 ± 7 | 73 ± 11 | $107^{+53}_{-45}$ |
| **Average** | -- | -- | -- | -- | 65 ± 6 | $143^{+33}_{-29}$ |

The mean concentrations of $Na^+$, $K^+$, $Mg^{2+}$, $Cl^-$, $NO_3^-$, and $SO_4^{2-}$ in the samples are 34.9, 9.2, 11.1, 84.0, 44.4, and 94.5 ng·g$^{-1}$, respectively, which are similar to the values of surface snow samples collected along the Chinese inland Antarctica traverse route, about ~60km from the study site (Shi et al., 2021). The mean $\delta^{18}O$ and $\delta^2H$ of the blue ice are -40.3‰ and -321.2‰, respectively, also similar to those of the nearby surface snow (Ma et al., 2020). These stable water isotope values are much higher than those of the snow and ice on the East Antarctic plateau today (Xiao et al., 2008 and references therein) and during the Last Interglacial period (e.g., Petit et al., 1999). Assuming no isotopic modifications after snow deposition and during ice flow, the original deposition site of the surface ice at the Grove Mountains BIA today was likely local, but we acknowledge that the exact location remains unknown. Any future attempt





to interpret those isotope data in the context of paleoclimate would require a more thorough investigation of the provenance of the blue ice.

170

## 4. Discussion and Conclusions

That the surface ice at Grove Mountains BIA dates back to the Last Interglacial has important implications for paleoclimate studies. First, there are no previously published deep ice core records from the Princess Elizabeth Land that date back to the Last Interglacial. Consequently, the Grove Mountains BIA holds the potentials to provide large-volume ice samples to study the climate variations during the Last Interglacial in the Indian Ocean sector of Antarctica. Second, in the Allan Hills BIA, where ice flows towards and overrides subglacial mountains, a layer of ice near the bedrock dates back beyond 2 Myr (Yan et al., 2019). The glaciological similarities make it possible that ice underneath the Grove Mountains BIAs is older than 800 kyr as well. Finally, in addition to retrieving ice cores, a synergistic effect of drilling operations in the Grove Mountains is the potential recovery of bedrock samples. Some previous studies suggest the East Antarctic Ice Sheet would retreat beyond Grove Mountains during past warm intervals (Liu et al., 2010). Bedrock samples from the Grove Mountains can help evaluate this hypothesis and improve our understanding of ice sheet behaviors in a fast-warming world.

The mean age of two surface ice samples from the Grove Mountains BIA, dated radiometrically by $^{81}$Kr, is $143^{+33}_{-29}$ kyr. Furthermore, a meteorite (GRV021710) that has a terrestrial age of 260 kyr suggests that Grove Mountains BIA could harbor even older ice. In general, the major chemical ions and stable water isotopes of the ice resemble those of the nearby surface snow and differ from those from Antarctic plateau sites, suggesting that the blue ice is originated nearby. In addition, the bedrock at the Grove Mountains BIA could reveal important information about the stability of the East Antarctic Ice Sheet in past warm intervals. Given these considerations, we conclude that Grove Mountains are a region with high scientific values. Future drilling operations in the Grove Mountains BIA could also benefit from its close proximity to a nearby Antarctic research base (Chinese Taishan Station; Figure 1a). More systematic glaciological surveys in this region are called for, with the ultimate goal of retrieving ice cores and bedrock samples to study paleoclimate and past ice sheet behaviors.

## Conflict of Interest

The authors declare no conflicts of interest relevant to this study.



**Credit authorship contribution statement**

GS and ZH conceived the study. YY, WJ, and GS designed and wrote the manuscript with the

support of all co-authors. ZH and GS analyzed ions and stable water isotopes. WJ, FR, GY,

and ZL analyzed the Kr data. YH, XT, LL, and GS prepared the figure.

**Data Availability Statement**

Data presented in this work are included in the main text.

**Acknowledgments**

This work was supported by the National Science Foundation of China (Grant Nos.

42276243, 41922046, 42071306, T2325024, and 41727901), the Innovation Program for

Quantum Science and Technology (2021ZD0303101), and the Program of Shanghai

Academic/Technology Research Leader (Grant No. 20XD1421600). The authors are grateful

to CHINARE members for their support and assistance in sample collection.

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
