# Peer review of "Brief communication: Identification of 140,000-year-old blue ice in Grove Mountains, East Antarctica, by krypton-81 dating"

_EGUsphere, 2023_

## Author Comment (AC1)

**Referee #1**

We thank you very much for the thoughtful and thorough review of our manuscript. The very helpful comments and suggestions have greatly improved the quality of this paper. Below, we give a point-by-point response to the comments and suggestions of the reviewer, in the order of (1) comments from Referees, (2) author's response (referee comments in black; author's response in blue).

**(1) comments from Referees**

General comments

In the manuscript "Brief communication: Identification of 140,000-year-old blue ice in Grove Mountains, East Antarctica, by krypton-81 dating" by Z. Hu et al., the authors present the results of the determination of the age of air bubbles in blue ice from the Grove Mountains BIA in Elizabeth Princess Land, East Antarctica, using the 81Kr technique. Two samples showed an average age of ~143 ka. Together with the reported terrestrial age of a chondrite, the authors concluded that Grove Mountains BIA holds considerable potential for paleoclimate studies.

The manuscript introduces the identification of a novel area of BIA in Antarctica that could be suitable for paleoclimate studies. The authors highlight that the region of Grove Mountains can be included in the record of BIAs in Antarctica such as Mount Moulton and Patriot Hills in West Antarctica, and Allan Hills, Taylor Glacier, and Larsen Glacier in East Antarctica, holding old ice which could help to investigate past climate changes.

The text is generally clear and the enclosed results are worthy of publication, however, I do suggest some improvements, especially regarding the identification of Grove Mountains as a potential ice coring/paleoclimate studies site. The discussion should be expanded on this topic, including some comments on radar profiles and ice stratigraphy. The dating technique is properly described and I do not have specific comments on that. The use of the English language is appropriate and the concepts are clearly presented.

Here I enclose some comments that I hope would improve the quality of the manuscript. I am looking forward to reading the new version of the text.

**(1) author's response**

Thanks for your efforts, and we appreciate your kind words about the value of this work.

**(2) comments from Referees**

Specific comments:

Materials and methods section: the authors highlight since the introduction section that the BIA of Grove Mountains (GM) could hold ice hold enough for paleoclimate studies, especially for ice core drilling. I believe that this strong affirmation should be followed by a more expanded discussion in the "site and sample" subsection. In Figure 1c the authors show a radar profile of the transect, the discussion may start from this picture such as: how is the ice layering there? Where potential drilling sites could be identified? Is it an area where short cores or long cores could be drilled? If the whole point of this manuscript is to identify a new area for ice coring and paleoclimate studies, this additional discussion would improve the impact of this study.

**(2) author's response**

Agree and thank you for the thoughtful comments. Following your comments, additional discussion about the shallow ice core drill sites was included in section 4.0, as follows,

*To obtain old ice, the potential drilling sites in the BIAs are usually located in the upstream of the ice flow, where the ice stream was blocked by the nunataks for the first time, like the drilling sites in Allan Hills (Yan et al., 2019). Accordingly, the potential old ice drilling sites in Grove Mountains BIA are expected to be around the mid-Gale Escarpment, following the ice flow direction in this region (Figure 1). It is noted that there are a large number of ice crevasses formed on the side of the mid-Gale Escarpment facing the ice flow, making it currently inaccessible from the ground. The radar profile provides the direct observations of deep englacial stratigraphy in this BIA (Figure 1c). However, only some disturbed layers can be imaged, at depths of 200-400m beneath the ice surface. The radar image showed that the internal layers are not well identified at a depth of >500m, which could be the result of complex ice flow patterns around the nunataks. Nevertheless, it can be seen that the subglacial topographic mountains may cause the ice to flow toward the surface, especially near the nunataks, where the ice depth is relatively shallow, at a few hundred meters. These areas are expected to be potential shallow ice core drilling sites, i.e., easier access to the bottom old ice.*

In addition, the detailed methods on the ice radar survey were included in the Materials and Methods, subsection 2.3 Ice radar survey.

**(3) comments from Referees**

Materials and methods section: around line 80 the authors describe how the samples were collected. Could you please provide some additional information? What was the size of the samples? Depth, length, etc.. Did you perform a preliminary visual inspection of the ice? Any cracks, or melted layers? Do you have pictures of the sampling in Antarctica and of the samples? A picture of the sampling in the field would be very interesting to see.

**(3) author's response**

Thanks for your useful comments. More information on the sampling is included in this section, as follows:

*The sampling size is approximately 40 × 40 × 40 cm. The ice samples are irregular blocks with length ranging from about 10 cm to 30 cm. Based on the visual inspection, there are no clear cracks and melted layers in the ice at the sampling site.*

In addition, one picture showing the surface morphology of the blue ice at the sampling sit is included in Figure 1 (Figure 1d). Accordingly, this subsection (2.1 Site and sample description) was re-visited.

**(4) comments from Referees**

Discussion and conclusion section: In line 175 the authors stress that the GM BIA could provide large ice samples to study past climate changes. To improve the discussion and support the conclusions Hu et al. here can include some lines about stratigraphy and potential ice core drilling discussed in the "material and methods section" (see comment above).

**(4) author's response**

Agree and thanks.
Additional discussion on the stratigraphy and potential ice core drilling sites is now included in the section 4.0 **Discussion and Conclusions.**

**(5) comments from Referees**

Technical corrections

Line 35: would be worth mentioning the Beyond Epica Oldest Ice project and include a more recent reference since the drilling has already started.
See        https://tc.copernicus.org/articles/12/2167/2018/        and        the        official        website
https://www.beyondepica.eu/en/about/

**(5) author's response**
Thank you for bringing this point, added in the revised manuscript. Now it reads,
*International efforts are underway to locate and retrieve an ice core dating back to 1.5 Myr that is in stratigraphic order (Fischer et al., 2013; Passalacqua et al., 2018), e.g., the ongoing Beyond Epica Oldest Ice project at Little Dome C (https://www.beyondepica.eu/en/about/).*

**(6) comments from Referees**

Line 40: please change "being exhumed" to "brought to the surface"

**(6) author's response**
Done.

**(7) comments from Referees**

Lines 41-42: the affirmation "The presence of meteorites that have terrestrial ages up to 2 Myr in the BIAs hints at the existence of ice that is older than 800 kyr (Scherer et al., 1997)" is a bit strong especially because Scherer at al. dated the ice close the meteorite to be 325 ka old. The sentence could be rephrased with "300 ka old ice".

**(7) author's response**

Agree and done, now it reads,

*The presence of meteorites that have terrestrial ages up to 2 Myr in the BIAs hints at the existence of 300 kyr old ice (Scherer et al., 1997).*

**(8) comments from Referees**

Lines 45: please substitute "Nevertheless, to date debris-free ice samples have been recovered from only five blue ice areas for the purpose of paleoclimate studies" with "To date debris-free ice, ice samples were recovered from only five blue ice areas" and remove "for the purpose of paleoclimate studies"

**(8) author's response**

We have rephrased this sentence to improve clarity: Debris-free ice samples were recovered from only five blue ice areas for the purpose of paleoclimate studies so far. "To date" actually means "so far" in this context, and might be confusing to readers as it can also be interpreted as "to determine the age of".

**(9) comments from Referees**

Line 49: please substitute the sentence with "In this study, we focus on the Grove Mountains, which consist of a series…"

**(9) author's response**

Done.

**(10) comments from Referees**

Figure 1b: please enlarge the red triangle size to be easier to spot.

**(10) author's response**

Done.

**(11) comments from Referees**

Line 172: Please start the sentence with "Our study shows that ice surface at Grove Mountains dates back

to the Last Interglacial, holding important ..”

**(11) author's response**
Revised, and now reads,
*Our study shows that ice surface at Grove Mountains dates back to the Last Interglacial, holding important implications for paleoclimate studies.*

**(12) comments from Referees**

Lines 177-180: this is a very "heavy" sentence and I suggest that this might be too much in this manuscript. Then Allan Hills is located in West Antarctica, which may be too far to make a comparison. I suggest avoiding this sentence and focusing only on the obtained results for GMs and suggesting the presence of older ice.

**(12) author's response**
Allan Hills is also located in East Antarctica, as Grove Mountain does. Nevertheless, we acknowledge that a direct analogy about the presence of old ice needs more information to corroborate. Therefore, this sentence was removed in the updated version. This paragraph was re-organized. Please refer the first paragraph in section 4. Discussion and Conclusions.

**(13) comments from Referees**

Lines 185-190: I suggest moving those lines after line 173 to make this section more easy to read. It is easier for the reader to be reminded of the results at the beginning of the results section.

**(13) author's response**
Done.

**(14) comments from Referees**

Lines 173 onwards should be then adjusted consequently.

**(14) author's response**
This paragraph was re-organized.

**(15) comments from Referees**

Line 191: please substitute "past warm intervals" with "past Interglacials"

**(15) author's response**
Done.

**References**

Fischer H., Severinghaus J., Brook E., Wolff E., Albert M., Alemany O., Arthern R., Bentley C., Blankenship D., Chappellaz J., Creyts T., Dahl-Jensen D., Dinn M., Frezzotti M., Fujita S., Gallee H., Hindmarsh R., Hudspeth D., Jugie G., Kawamura K., Lipenkov V., Miller H., Mulvaney R., Parrenin F., Pattyn F., Ritz C., Schwander J., Steinhage D., van Ommen T. and Wilhelms F. (2013) Where to find 1.5 million yr old ice for the IPICS "Oldest-Ice" ice core. *Clim. Past* 9, 2489-2505.

Passalacqua O., Cavitte M., Gagliardini O., Gillet-Chaulet F., Parrenin F., Ritz C. and Young D. (2018) Brief communication: Candidate sites of 1.5 Myr old ice 37 km southwest of the Dome C summit, East Antarctica. *The Cryosphere* 12, 2167-2174.

Scherer P., Schultz L., Neupert U., Knauer M., Neumann S., Leya I., Michel R., Mokos J., Lipschutz M.E., Metzler K., Suter M. and Kubik P.W. (1997) Allan Hills 88019: An Antarctic H‑chondrite with a very long terrestrial age. *Meteorit. Planet. Sci.* 32, 769-773.

Yan Y., Bender M.L., Brook E.J., Clifford H.M., Kemeny P.C., Kurbatov A.V., Mackay S., Mayewski P.A., Ng J., Severinghaus J.P. and Higgins J.A. (2019) Two-million-year-old snapshots of atmospheric gases from Antarctic ice. *Nature* 574, 663-666.

**End of responses to Referee #1.**

---

## Author Comment (AC2)

**Reviewer #2**

**We are very grateful to you for the detailed comments and very useful suggestions. The manuscript has been modified based on these comments/suggestions. Below, we give a point-by-point response to the comments and suggestions, in the order of (1) comments from Referees, (2) author's response.**
**Comments are in black, and the responses are in blue.**

**(1) comments from Referees**

This is an extremely simple brief communication whose purpose is to show that the grove mountains blue ice area would be a good place to do ice coring for palaeoclimate. The novelty in the paper consists of two (similar) dates obtained using 81Kr dating, which show that ice 140000 years old is present at the surface. The authors comment that the isotopic and chemical data suggest that the ice originated somewhere relatively local and not high on the East Antarctic plateau.

There is not really much to comment on. The date is hard-won, as this is a dating method that requires a lot of effort and a lot of ice. The finding is novel if a bit limited - it would of course have been much more interesting if the authors had been able to say anything about ice at depth, or to demonstrate that the greenhouse gas content of the ice was compatible with the ages they found. However, it is what it is. This seems like a result and proposal that is worth publicising, and the science in the paper is correct. I therefore recommend publication with very limited revision.

**(1) author's response**

We thank you for your time in reading our manuscript and the very constructive comments. We totally agree with you that the greenhouse gas data would be more helpful in terms of assessing the values of Grove Mountain BIA as paleoclimate archives. In addition, a deeper ice core will provide more information about the stratigraphy, which could be constrained by gas measurements as well. Unfortunately, these surface ice samples were not analyzed for the greenhouse gas content. In the revised manuscript, we included a discussion on the possible ice core drilling sites in Grove Mountains. In the future work of the blue ice core, we will include the greenhouse gas analysis. Thank you very much for the suggestions.

**(2) comments from Referees**

Minor comments:

Line 25: Princess Elizabeth rather than Elizabeth princess (it's right elsewhere in the paper)

**(2) author's response**
Corrected, thanks.

**(3) comments from Referees**

Line 53 "potential" rather than "potentials"

**(3) author's response**
Corrected.

**(4) comments from Referees**

Table 1. For Allan Hills it might be nice to add a more recent paper such as Yan 2023 (Yan, Y., Kurbatov, A. V., Mayewski, P. A., Shackleton, S., and Higgins, J. A.: Early Pleistocene East Antarctic temperature in phase with local insolation, Nature Geoscience, 16, 50-55, doi: 10.1038/s41561-022-01095-x, 2023)

**(4) author's response**
Thank you for the suggestion. However, the Yan et al (2023) paper deals with existing data first reported in earlier publications. The Yan et al (2019), cited in the first draft, is the first paper that reports the existence of old ice (>2 Ma) in the Allan Hills BIAs.

**(5) comments from Referees**

On page 5 (around line 108), the reader would appreciate more information about why 85Kr indicates contamination. I suppose giving its short half life would explain this.

**(5) author's response**
We follow the reviewer's suggestion and added the following sentence to make it clearer. Now it reads (subsection 2.2),
"*The anthropogenic $^{85}Kr$ isotope is analyzed since it has a half-life of 10.7 years, making it a good indicator of cross-sample contamination from the modern reference sample.*"

**(6) comments from Referees**

Line 161. If possible it would be nice to see a small table with the water isotope data for Grove Mts, for nearby sites from the paper by Ma et al, and for inland sites (Dome A, Vostok, Dome Fuji), along with their elevations. This would help to make the point that the ice must have a local rather than plateau origin.

**(6) author's response**
Agree and thank you for this helpful comment. For the Brief Communication in The Cryosphere, the total number of tables and figures should be no more than three. In the main manuscript, we have two tables and one figure; thus, we included this table in the supplementary materials (Table S1).

Table S1 Stable isotopes of water in the snow and ice from different sites in Antarctica

| Sites | $\delta^{18}O(H_2O)$ /‰ | $\delta^2H$ $(H_2O)$/‰ | Elevation/m | References |
|---|---|---|---|---|
| Blue ice in Grove Mountains | -40.3 | -321.2 | ~2000 | This study |
| Surface snow near Grove Mountains [a] | -37.0±1.6 | -289±14.8 | 2556 | (Ma et al., 2020a) |
| Dome A | -58.5±2.3 | -449.4±17.0 | 4089 | (Ma et al., 2020b) |
| Dome A | -58.4 | -450 | 4093 | (Xiao et al., 2008) |
| Dome C | -50.1 | -390 | 3240 | (Stenni et al., 2001) |
| Vostok | -56.4 | -440 | 3490 | (Ekaykin et al., 2004) |
| Dome Fuji | -54.9 | -425 | 3810 | (Watanabe et al., 2003) |
| Dome B | -55.2 | -430 | 3650 | (Masson et al., 2000) |

[a] Surface snow samples were collected on the Chinese inland Antarctic expedition traverse route from Zhongshan to Dome A (Figure 1a), about 65 km from the Grove Mountains.

**References**

Ekaykin, A.A., Lipenkov, V.Y., Kuzmina, I.N., Petit, J.R., Masson-Delmotte, V., Johnsen, S.J., 2004. The changes in isotope composition and accumulation of snow at Vostok station, East Antarctica, over the past 200 years. Ann. Glaciol. 39, 569-575.

Ma, T., Li, L., Li, Y., An, C., Yu, J., Ma, H., Jiang, S., Shi, G., 2020a. Stable isotopic composition in snowpack along the traverse from a coastal location to Dome A (East Antarctica): Results from observations and numerical modeling. Polar Sci. 24, 100510.

Ma, T., Li, L., Shi, G., Li, Y., 2020b. Acquisition of Post-Depositional Effects on Stable Isotopes ($\delta^{18}O$ and δD) of Snow and Firn at Dome A, East Antarctica. Water 12, 1707.

Masson, V., Vimeux, F., Jouzel, J., Morgan, V., Delmotte, M., Ciais, P., Hammer, C., Johnsen, S., Lipenkov, V.Y., Mosley-Thompson, E., 2000. Holocene climate variability in Antarctica based on 11 ice-core isotopic records. Quaternary Res. 54, 348-358.

Stenni, B., Masson-Delmotte, V., Johnsen, S., Jouzel, J., Longinelli, A., Monnin, E., Röthlisberger, R., Selmo, E., 2001. An oceanic cold reversal during the last deglaciation. Science 293, 2074-2077.

Watanabe, O., Kamiyama, K., Motoyama, H., Fujii, Y., Igarashi, M., Furukawa, T., Goto-Azuma, K., Saito, T., Kanamori, S., Kanamori, N., 2003. General tendencies of stable isotopes and major chemical constituents of the Dome Fuji deep ice core. Memoirs of National Institute of Polar Research Special Issue No.57. Tokyo: National Institute of Polar Research,.

Xiao, C., Li, Y., Hou, S., Allison, I., Bian, L., Ren, J., 2008. Preliminary evidence indicating Dome A (Antarctica) satisfying preconditions for drilling the oldest ice core. Chin. Sci. Bull. 53, 102-106.

**End of responses to Reviewer #2.**

---

## Author Response (AR2)

**Response to the editor**

We appreciate Prof. Chris R Stokes (the handling editor) for the time in processing and reviewing our manuscript. Below, we give a point-by-point response to the comments and suggestions of the editor, in the order of (1) comments from editor, and (2) author's response (editor comments in black; author's response in blue).

**Editor's comments:**

Dear Guitao and co-authors,

Thank you for your prompt and clear response to the reviewer comments. You have addressed their very minor comments and I'm delighted to accept the manuscript, subject to some technical corrections detailed below (line numbers refer to version with changes tracked):

Line 47: This sentence may be clearer as: "Thus far, debris-free [...] for the purpose of paleoclimate studies (Table 1 and Figure 1): Mount...."
**Author's response**
Corrected.

Line 90: "Based on visual inspection, there are..."
**Author's response**
Corrected.

Line 198: "...suggesting that the blue ice originated nearby."
**Author's response**
Corrected.

Line 203:"...located upstream of the ice flow, where the ice stream was blocked by the nunataks, similar to the drilling sites..."
**Author's response**
Corrected.

Line 205: "...in the Grove Mountains BIA..."
**Author's response**
Corrected.

Line 215:"...i.e., easier access to the oldest ice."
**Author's response**
Corrected.